# Coarse-Grained Simulations of Release of Drugs Housed in Flexible Nanogels: New Insights into Kinetic Parameters

**DOI:** 10.3390/polym14214760

**Published:** 2022-11-07

**Authors:** Manuel Quesada-Pérez, Luis Pérez-Mas, David Carrizo-Tejero, José-Alberto Maroto-Centeno, María del Mar Ramos-Tejada, Alberto Martín-Molina

**Affiliations:** 1Departamento de Física, Escuela Politécnica Superior de Linares, Universidad de Jaén, Linares, 23700 Jaén, Spain; 2Departamento de Física Aplicada, Universidad de Granada, Campus de Fuentenueva s/n, 18071 Granada, Spain; 3Instituto Carlos I de Física Teórica y Computacional, Universidad de Granada, Campus de Fuentenueva s/n, 18071 Granada, Spain

**Keywords:** drug delivery, controlled release, nanogel, coarse-grained simulation

## Abstract

The diffusion-controlled release of drugs housed in flexible nanogels has been simulated with the help of a coarse-grained model that explicitly considers polymer chains. In these in silico experiments, the effect of its flexibility is assessed by comparing it with data obtained for a rigid nanogel with the same volume fraction and topology. Our results show that the initial distribution of the drug can exert a great influence on the release kinetics. This work also reveals that certain surface phenomena driven by steric interactions can lead to apparently counterintuitive behaviors. Such phenomena are not usually included in many theoretical treatments used for the analysis of experimental release kinetics. Therefore, one should be very careful in drawing conclusions from these formalisms. In fact, our results suggest that the interpretation of drug release curves in terms of kinetic exponents (obtained from the Ritger–Peppas Equation) is a tricky question. However, such curves can provide a first estimate of the drug diffusion coefficient.

## 1. Introduction

In the last decades, nanogels have attracted growing interest due to their potential application as drug delivery vehicles [1,2,3,4]. Among their most attractive characteristics, researchers have found the following ones: (i) their size allows them to traverse capillaries and penetrate tissues; (ii) they can be designed to respond to specific stimuli; (iii) nanogels can encapsulate high amounts of drugs and/or biologically active macromolecules and release them in a controlled manner; (iv) nanogels are easy to synthesize on an industrial scale.

One of the main goals of drug delivery is controlled release. That is why many studies on nanogels as drug carriers pay special attention to release kinetics. Some examples of drugs whose release kinetics have been studied are: aspirin [5], doxorubicin (anticancer drug) [6,7,8], ethosuximide (antiepileptic) [9], insulin [10], terfenadine (antihistamine) [11], 5-fluorouracil (anticancer drug) [11], and ginsenoside CK (anti-inflammatory and antitumor drug) [12]. The conclusions of some of these works are disparate and even contradictory. For instance, Cazares-Cortés et al. have concluded that the drug is more rapidly released when the nanogel shrinks [7]. In contrast, Álvarez-Bautista et al. have claimed that the release kinetics speeds up if the nanogel swells [11]. Surprisingly, Aguirre et al. found that the effect of the nanogel size on the kinetics is negligible [8]. Apart from this disparity, it should be mentioned that the dynamic dialysis method has extensively been used to measure the release kinetics, but the diffusion of the drug through the dialysis membrane could delay its appearance in the sampling compartment. For that reason, the reliability of this method for measuring release kinetics has been questioned [13].

In any case, controlled release relies on the precise knowledge of the mechanisms involved in this process. In relation to this, such processes are classified into three main categories: diffusion-controlled, swelling-controlled, and chemically controlled release [14]. This is just an ideal classification because, in practice, a combination of these mechanisms takes place. In fact, diffusion is always present in drug delivery.

The kinetics of diffusion-controlled release can be mathematically predicted from solutions of Fick’s second law as long as the diffusion coefficient of the drug inside the polymer network, $D_{g}$, is precisely known. This is not a trivial task. In addition, it should be mentioned that such solutions are usually derived under the assumption of perfect sink conditions. This means that drug molecules (which we will also refer to as the solute) are removed as soon as they reach the border of the matrix. For gels, this can be achieved by stirring, but this method is not useful in the case of nanogels.

Alternatively, solute release from nanogels and other nanocarriers can also be simulated [15,16]. In fact, diffusion-controlled release processes from spherical matrices were initially simulated through lattice models [17,18,19]. It should be pointed out, however, that the inner structure of the polymer network was not explicitly considered. In addition, the concentration of the drug outside the polymer network remained null. In other words, perfect sink conditions were assumed.

Recently, Maroto-Centeno and Quesada-Pérez have also simulated drug release from a rigid gel of nanometric size [20]. They did explicitly consider the polymeric chains as well as the monomeric units and crosslinkers constituting them through a coarse-grained (CG) model, which had been previously applied to obtain helpful information about several single-particle properties (swelling, mass distribution, ionic distributions, effective charge, and electrostatic potential) without requiring details on the chemical nature [21,22,23,24]. The algorithm employed therein only requires the free diffusion coefficient, which is much easier to estimate or measure than $D_{g}$. In addition, these simulations were not performed under the assumption of perfect sink conditions. Maroto-Centeno and Quesada-Pérez reported that the kinetic exponent characterizing the first 60% of the release curve deviates from the classical prediction for spherical matrices if the solute diameter is high enough.

As mentioned before, the simulations performed by these authors were restricted to a rigid nanogel. In practice, however, many polymeric nanogels are flexible, and this allows them to respond to external stimuli. In addition, flexibility facilitates the diffusion of solute particles and even promotes their movement through the polymer network when the particle size is comparable to the mesh size or larger [25,26,27,28]. Consequently, the main goal of this work is to extend this preliminary work to flexible nanogels. We also want to find out to what extent chain flexibility facilitates the release of drug molecules loaded inside nanogels and modifies the corresponding kinetics. According to some recently published results, the diffusion coefficient in flexible polymer networks can be significantly greater than the value corresponding to rigid networks for moderate polymer volume fractions and solute diameters [29]. Thus, we wonder if the effect of flexibility on the release kinetics can be so significant.

Many experimental drug release curves are analyzed in terms of a kinetic exponent obtained by fitting such curves to a power law (the Ritger–Peppas Equation) [30]. The effect of flexibility on this kinetic parameter is also studied here. Our in silico experiments suggest that the interpretation of this exponent is not a simple matter since surface phenomena or the initial drug distribution can significantly alter its value.

## 2. Model and Simulations

In this work, the coarse-grained model of nanogel has been employed. This model has been successfully applied to the study of single-particle properties of nanogels and the interaction between them [21,22,23,24]. In addition, GC models allow considering different polymer-solute interactions or polydispersity in polymer gels [31,32,33]. According to this simplified representation of reality, monomeric units of the polymeric network and drug molecules were modeled as spheres, whereas the solvent was considered a continuum. The diameter of the monomeric units and the crosslinker molecules that join polymeric chains was 0.65 nm. Many real monomers have diameters close to this value [34]. The diameter of the drug molecule varied from 0.75 to 1.75 nm. Many drug molecules have mean diameters in this range (e.g., doxorubicin [35]). The nanogel was made of 692 polymer chains of 4 monomeric units connected by 404 crosslinkers. A high crosslinker-to-monomer ratio was deliberately chosen to generate a nanogel with a polymer volume fraction (φ) close to 0.08. According to a previous work, the diffusion coefficients in rigid and flexible gels significantly differ precisely from volume fractions of this order [29]. We would like to find out to what extent such differences affect the release kinetics.

Before simulating the drug release process itself, a nanogel with these characteristics was generated from a simulation procedure described in a previous paper [36]. Figure 1 displays a cross-section of this nanogel passing through its center of mass. This figure was made from the positions of the monomeric units and crosslinkers just at the end of the simulation to generate the nanogel. As can be seen, the crosslinkers exhibit some degree of spatial ordering: the mean distance between them is 2.2 nm. Figure 1 also shows that the nanogel does not have a well-defined surface, but a mean geometrical radius ($R_{ng}$) can be estimated from its radius of gyration [37]. It should be mentioned, however, that there are a few monomeric units (about 10%) beyond the border that defines the imaginary sphere with this geometrical radius. The geometrical diameter of the nanogel used here ($2R_{ng}$) turned out to be 21.3 ± 0.2 nm. A mean polymer volume fraction of 0.081 was also estimated. In our case, this polymer volume fraction corresponds to a highly crosslinked nanogel, but it might also be representative of moderately collapsed nanogels with a lower degree of crosslinking.

Before starting the release process, 100 drug particles (also drawn in Figure 1) were randomly placed in the voids of the nanogel. The movements of the different particles of the system during the release were executed following Brownian dynamics methods. Solute particles moved according to the Cichocki–Hinsen algorithm [38]. This stochastic procedure was proposed to consider the case of suspensions of hard spheres and has been employed to simulate the diffusion of spherical particles in physical chemistry and biology [39,40,41,42]. The idea behind this algorithm is quite simple. In every simulation step, the particles move as if diffusing freely. However, when they run into an obstacle (such as polymer chains), such movements are rejected. These rejections reduce the mean square displacement and the diffusion coefficient. Accordingly, the Cartesian components of the displacement vector of particle i during a time step Δt are given by [38]:(1)$$\Delta r_{mi}=\sqrt{2D_{0i}\Delta t}N\left(0,1\right)$$
where m stands for the spatial directions (x, y, or z), $D_{0i}$ is the free diffusion coefficient of the particle in the solvent and N(0,1) is a random number generated according to a Gaussian distribution of zero mean and unit standard deviation. The free diffusion coefficient of particle i was estimated from the Einstein–Stokes relationship, $D_{0i}=k_{B}T/6\pi \eta R_{i}$, where $k_{B}$ is Boltzmann’s constant, T is the absolute temperature, η stands for the solvent viscosity, and $R_{i}$ is the radius of the particle. In any case, the displacement vector whose components are given by Equation (1) is only tentative. This displacement will be rejected if two particles overlap. In this way, steric (excluded-volume) interactions are considered in the Cichocki–Hinsen algorithm.

In the case of rigid nanogels, monomeric units and crosslinkers remain fixed, but they move in flexible nanogels. Therefore, the elastic forces between bonded particles must also be considered. In these simulations, elastic forces satisfy Hooke’s law:(2)$${\vec{F}}_{ij}=-k_{e}\left(r_{ij}-r_{0}\right){\hat{r}}_{ij}$$
where ${\vec{F}}_{ij}$ is the force that particle i exerts on particle j, $k_{e}$ is the elastic constant, $r_{ij}$ is the distance between particles i and j,$\;r_{0}$ is the bond length at equilibrium and ${\hat{r}}_{ij}$ is the unit vector pointing from particle i to particle j. As in previous works [43], $k_{e}=0.4$ N/m and $r_{0}=0.65$ nm (the diameter of monomeric units).

Elastic forces were also included in the algorithm through Brownian dynamics. In the absence of hydrodynamic interactions, the Cartesian components of the displacement vector of monomeric units and crosslinkers are given by [44]:(3)$$\Delta r_{mi}=\sqrt{2D_{0i}\Delta t}N\left(0,1\right)+D_{0i}F_{mi}\Delta t/k_{B}T$$
where $F_{mi}$ is the Cartesian m-component of the net force exerted on particle i. As in the case of solute particles, this provisional displacement will not be accepted if it produces overlaps.

The time step should be small enough to guarantee the insensitivity of the results to this parameter and to avoid instabilities. On the other hand, very small time steps lead to extremely time-consuming simulations. After some preliminary research, a time step of 3·10^−12^ s was chosen. Simulations were performed at 293 K. A cubic simulation box with a side of 10,000 nm was employed. In a given simulation, each release process was repeated 18 times for statistical averaging. In turn, each simulation was repeated three times to obtain independent fitting parameters. Consequently, 54 iterations of the release process were employed to obtain kinetic parameters and their corresponding uncertainties. The programs to generate the initial configuration of the nanogel and simulate drug release were homemade and written in C. These programs will be available on request to interested readers as soon as user guides are ready.

## 3. Results and Discussion

### 3.1. Drug Distributed throughout the Nanogel

In the first series of simulations, the drug molecules were initially placed in the voids located at a distance less than $R_{ng}$ from the center of mass of the nanogel (see Figure 1). In other words, the solute particles were distributed throughout the nanogel. This assumption is common in many models based on solutions of Fick’s second law.

The main output of our simulations is the fraction of drug released (f) as a function of time (t). We consider that solute particles leave the nanogel when the distance between them and the center of mass of the polymer network is greater than the radius of the nanogel. When this occurs, the number of particles that have left the nanogel increases by one unit, and f grows. Figure 2 shows this parameter as a function of time for five solute diameters. The drug was housed in a flexible nanogel. In neutral gels, the diffusion coefficient decreases with solute size. Consequently, the fraction of drug released is expected to decrease with this parameter. However, this series of functions does not clearly show that behavior. In fact, at the initial stages of the process, the rate of release is greater for drug diameters of 1.75 and 1.50. Only after a certain time does the amount of released drug with diameters of 0.75, 1, and 1.25 nm exceed that of larger sizes.

In order to delve into the strange behavior of larger solute sizes, it is worth comparing with results obtained for rigid nanogels. Figure 3 shows the release kinetics obtained from simulations for drug molecules housed within rigid and flexible nanogels. In this case, the drug diameter is 1.75 nm.

This figure also includes the predictions computed from the solution of Fick’s second law for spherical gels under the assumption of perfect sink condition:(4)$$f=1-\frac{6}{\pi^{2}}\;\overset{\infty }{\underset{n=1}{\sum }}\frac{1}{n^{2}}\exp(-D_{g}n^{2}\pi^{2}t/R_{ng}^{2})$$
where $D_{g}$ is the diffusion coefficient inside the polymer network. This parameter was estimated from Equations (5) and (6) for rigid and flexible gels, respectively [45,46]:(5)$$D_{g}=D_{0}\exp\left(-0.84{\left(\varphi {\left(1+R_{s}/R_{p}\right)}^{2}\right)}^{1.09}\right)$$
(6)$$D_{g}=D_{0}\exp\left(-1.77{\left(\varphi {\left(1+R_{s}/R_{p}\right)}^{1.3}\right)}^{1.07}\right)$$
where $D_{0}$ is the free diffusion coefficient of drug molecules, $R_{s}$ is the solute (drug) radius, and *R_p_* is the polymer radius, which can be assumed to be close to the radius of the monomeric units in the case of polymeric chains.

At first glance, the large differences between the predictions of Equations (4)–(6) and the simulation results are striking. On the one hand, theoretical predictions show a gradual increase in the fraction of drug released. In addition, it can be seen that rigid nanogels release the drug more slowly than flexible ones, as expected.

On the other hand, the simulations show a very rapid release of 60% of the drug. From that point on, the process becomes much slower and even seems to reach a plateau (particularly in the case of rigid nanogels). Our simulations also reveal a counterintuitive behavior: release is faster for rigid nanogels in much of the process, particularly at the initial stages; only after a long time do flexible nanogels manage to release more drug than the rigid ones.

This assortment of puzzling findings could be justified as follows: the first drug molecules to leave the nanogel are those occupying the voids on its surface, and these molecules diffuse freely as they move out of the nanogel. However, when such molecules try to get into the polymer network, they run into obstacles. In other words, drug molecules on the surface preferentially diffuse outwards due to steric hindrances within. In addition, these drug molecules leave the nanogel in a short time, almost simultaneously, as can be concluded from the steep initial slope of the release curve shown in Figure 3. Thus, this surface anisotropic diffusion speeds up the drug release at the initial stages. At this point, it should be mentioned that the fraction of drug molecules initially located at the surface of the nanogel could be high. For example, let us consider a surface spherical shell whose thickness is 1 nm. Such a layer occupies more than 25% of the volume of the nanogel, but it can house 40% of 1-nm drug molecules (according to our simulations) because voids are more likely therein. Obviously, the presence of so many solute particles at the surface enhances the effects of anisotropic diffusion. Since Equation (4) does not take this surface effect into account, it is logical that its predictions underestimate the fraction of drug released. Figure 3 displays two cases in which such underestimation is noticeable. We should also keep in mind that this diffusive anisotropy is caused by steric interactions. Consequently, rigid gels should experience it to a greater extent, as seen in Figure 3. When the molecules on the surface have left the nanogel, the release of internal molecules begins. This process is controlled by conventional diffusion, which is faster in flexible gels. For this reason, these nanogels could release their load before the rigid ones in spite of being initially slower. However, in the case of moderate or high polymer volume fractions and/or large solute sizes, some drug molecules could be trapped for a long time if they do not find holes through which to escape. This would explain the extremely low release rates observed at long times in Figure 3.

In relation to Figure 2 and Figure 3, it is also worth mentioning that the time scale of the simulated release process is much smaller than that observed in release experiments performed with gels, microgels, and even nanogels. Several factors may help explain why the characteristic release times observed in simulations and experiments are so different. First, the reader should keep in mind that, in our simulations, the drug molecules must diffuse only a few nanometers before leaving the nanogel. It should also be noted that the only polymer–drug interaction included in these simulations is the steric one. The presence of attractive forces between the polymer chains and the solute could significantly slow down diffusion and almost immobilize the drug molecules [47]. Finally, it should not be forgotten that the dialysis membrane commonly used to monitor drug release over time could considerably distort the kinetics of this process by interposing a second barrier to solute diffusion [13].

It has long been known that the initial 60% of the release curve predicted by Equation (4) can be approximated by the Ritger–Peppas Equation [48]:(7)$$f=kt^{n}$$

In this power law, n=0.43 (for spherical gels) and [20]:(8)$$k≅2.246{\left(D_{g}/R_{ng}^{2}\right)}^{n}$$

Equation (7) is the starting point of many analyses of release kinetics, whose data are fitted using n and k as adjustable parameters [15,30]. According to Ritger and Peppas [48], n-values greater than 0.43 mean that the release is not purely diffusive.

Therefore, it is worth finding out to what extent the curves obtained by simulation obey this power law (Equation (7)) when the size of the solute changes. For each drug diameter, three independent release curves obtained from simulations were fitted to Equation (7). Figure 4 and Figure 5 display the mean values obtained for n and k, respectively, as a function of the drug diameter. Rigid and flexible nanogels were considered. Error bars stand for their standard deviation.

As can be seen in Figure 4, n decreases with the solute size. In order to correctly interpret this behavior, the reader should bear in mind that, at the initial stages of the release (t approaching zero), $t^{n}$ grows when n decreases. Consequently, the decreasing trend found for n means that the release kinetics speeds up with the drug diameter. This behavior is somewhat paradoxical since, within a gel, the diffusion of a neutral solute slows down as its size increases. The surface anisotropic diffusion driven by the steric interaction is responsible for this counterintuitive behavior again since excluded-volume effects grow with the size. This also implies that the effects of this phenomenon are attenuated by decreasing the drug diameter or the polymer volume fraction. For example, previous simulations performed with a rigid nanogel whose polymer volume fraction was 0.023 have shown that n does not deviate significantly from 0.43 for solute sizes less than 1.75 nm [20].

From Figure 4, one can also conclude that the deviations from the classical value are smaller for flexible gels. In fact, the n-value obtained for a diameter of 0.75 nm (the smallest studied here) matches the classical one. All of this is also consistent with the mechanism of surface anisotropic diffusion put forward here. In relation to Figure 4, it is also interesting to point out that some authors have previously reported *n*-values smaller than 0.43 for nanogels loaded with different antitumor drugs [8,49,50]. Figure 5 shows the k-values obtained from simulations for rigid and flexible nanogels. As can be seen, k is greater for flexible nanogels. This is logical since the diffusion coefficient is greater in these polymer networks. This figure also includes the k-values estimated from Equations (5) and (8) for rigid nanogels, and 8 and 6 for flexible polymer networks, using the n-values plotted in Figure 4 in both cases. In relation to Equation (8), we should keep in mind that, strictly speaking, it was derived assuming that n=0.43 [20]. However, Figure 4 reveals that other n-values are also possible for diffusion-controlled drug release. Here we have assumed that Equation (8) is also valid for these n-values. As can be concluded from Figure 5, the predictions of Equation (8) reproduce reasonably well the values provided by simulations. This supports the hypothesis that Equation (8) works for n≠0.43 and confirms that n and k are correlated. Until now, experimentalists have paid little attention to k. However, Figure 5 suggests that Equation (8) can provide a preliminary estimate of the diffusion coefficient $D_{g}$ from the values of n and k obtained by fitting reliable release kinetics. Recently, Ignacio and Slater have proposed a method that provides an estimate of the diffusion coefficient from a relaxation time of the release kinetics [51].

In relation to the fits performed with Equation (7), the $R^{2}$-coefficient (not shown) reveals that their statistical quality worsens with solute size and is somewhat worse for flexible nanogels. In this case, this coefficient even falls below 0.98 for the largest size. Better $R^{2}$-coefficients can be obtained if simulation data are fitted with the Weibull function (above 0.99 in all cases). However, this function involves parameters that no theory predicts.

### 3.2. Drug Distributed in the Core of the Nanogel

Nanogels must not only deliver the drug but also transport it to the pathological region. To do this, there must be some drug-polymer interaction that retains the drug within the nanogel during their journey. This interaction should be “switched off” by external stimuli when nanogels arrive at the desired place. For instance, charged drug molecules might be electrostatically trapped by pH-sensitive charged chemical groups. These forces could be switched off by changes in the pH. Regardless of the nature of this interaction, the binding of the drug is expected to be more intense at those points with a high local density of monomeric units.

Figure 6 displays the number density of monomeric units and crosslinker molecules at a distance r from the center of mass of the nanogel. As can be seen, the particles forming the polymer network are unevenly distributed. According to this figure, the nanogel can be split into two regions. In the inner region (the core), the density oscillates around certain high values. These oscillations reveal that the nanogel is structured in layers. The origin of such layers is the above-mentioned spatial ordering of the crosslinkers and the accumulation of monomeric units around them. In any case, the density drops very quickly to zero after a certain distance ($R_{core})$. We will refer to this outer region as the shell. In the presence of attractive polymer-solute interaction, the solute would be more weakly bound to the nanogel in the shell. As a limiting case, we will assume that only the core can retain drug molecules and deliver them at the desired time.

Consequently, we have also performed simulations in which solute particles are initially distributed in a sphere of radius $R_{core}\approx 9.1$ nm (see Figure 7). Figure 8 shows the fraction of drug released as a function of time for rigid and flexible nanogels loaded with drug particles whose diameter is 1.75 nm. The release kinetics provided by the simulations differ markedly from those shown in Figure 3. In this case, the release is much more gradual. These curves do not exhibit abrupt changes. In addition, the flexible nanogels release more drug than the rigid ones at all times, as would be expected. On the one hand, this shows that the distribution of the drug within the nanogel has a profound influence on the release kinetics. On the other hand, the comparison between Figure 3 and Figure 8 suggests that the effects of surface anisotropic diffusion reduce if the drug concentrates in the core of the polymer network.

Figure 8 also includes the theoretical predictions, but only as a reference for comparison. It should be mentioned that we are not dealing with uniform drug distribution, one of the basic assumptions used to derive Equation (4). The discrepancies between theory and simulations observed in Figure 8 are mainly quantitative: the theory overestimates the fraction of drug released. However, such differences cannot be exclusively attributed to the drug distribution since simulations were not performed under perfect sink conditions either.

One could be tempted to describe the release kinetics of Figure 8 in terms of the kinetic parameters n and k. It should be pointed out, however, that the release process does not begin until the solute housed in the core manages to pass through the nanogel shell, which involves some delay, $t_{d}$. Consequently, the release kinetics was fitted to this modified power law:(9)$$f=k{\left(t-t_{d}\right)}^{n}$$

In this case, the value of $t_{d}$ was straightforwardly determined from simulation data (f=0 for $t<t_{d}$). Therefore, the only adjustable parameters in Equation (9) are n and k. The $R^{2}$-coefficient of these fits turned out to be similar to those obtained previously. The quality of the fits is better for rigid gels and/or small solutes. Figure 9 shows the n-values obtained from these fits for rigid and flexible nanogels as a function of the solute diameter. It is quite enlightening to compare Figure 9 (drug housed just in the core) and Figure 4 (drug housed throughout the nanogel). As can be seen, n shifts to much higher values when the drug is housed in the core, which means that the release kinetics slows down. In fact, the values of this parameter are now above the classic value. According to Ritger and Peppas [48], n-values greater than 0.43 correspond to anomalous diffusion. In contrast, Figure 9 reveals that some purely diffusive release processes can be characterized by n>0.43. It should be pointed out that there is no contradiction between this statement and the interpretation of n reported by Ritger and Peppas because the starting hypotheses are different: n=0.43 for diffusive release was obtained under the assumptions of uniform distribution of the drug throughout the spherical polymeric matrix and perfect sink conditions. None of these hypotheses applies here. In any case, the comparison of Figure 4 and Figure 9 proves that the initial distribution of the drug has a very important effect on release kinetics.

Figure 10 displays the k-values obtained from the previous fits for drug molecules housed in the core of rigid and flexible nanogels as a function of the solute diameter. This figure also includes the predictions of Equations (5) and (8) for rigid nanogels and 8 and 6 for flexible polymer networks, using the n-values plotted in Figure 9. The main conclusion drawn from this figure is the suitable agreement between the k-values obtained by fitting the simulated drug release curves and the predictions computed from Equation (8) and the estimates of the diffusion coefficients. This again suggests that k is directly related to $D_{g}$ (see Equation (8)), which facilitates the physical interpretation of k.

Finally, it is quite instructive to assess the effect of flexibility at different solute sizes from the fraction of drug released at the longest time explored in our simulations (100 ns). Figure 11 displays this f-value as a function of the solute diameter for rigid and flexible nanogels. Drug molecules are housed in its core. The difference between them in this parameter is quite small for 0.75 nm but grows with the drug size. This obviously means that flexibility enhances the release kinetics, as expected.

## 4. Conclusions

In this work, we have simulated the release of a drug loaded in rigid and flexible nanogels using a coarse-grained model that considers the polymer chains explicitly. This model allows us to simulate and better understand diffusion-controlled release processes whose time scale is very small. Furthermore, the algorithm used here does not require knowledge of the diffusion coefficient within the nanogel, which is not easy to measure or calculate.

When the drug is loaded throughout the nanogel, the simulated release curves noticeably differ from theoretical predictions and even exhibit behaviors challenging our reasoning. For example, during much of the process, rigid nanogels can release drug faster than flexible ones. The anisotropic diffusion of the drug loaded in the voids on the nanogel surface is largely responsible for these remarkable behaviors. Anisotropic diffusion also explains why the kinetic exponent characterizing the first 60% of the release curve is smaller than the classical value and decreases with the solute size. It should also be mentioned that diffusive anisotropy is a surface phenomenon. Consequently, it will be more important for systems with high surface-to-volume ratios, such as nanogels.

The effects of surface anisotropic diffusion will be smaller if: (i) the size of the gel grows; (ii) the drug size decreases; (iii) the drug is loaded only in its core. In the latter case, the particles must diffuse through the nanogel shell before leaving it. This prevents a massive release, and a better qualitative agreement between theory and simulation is achieved. Our results, therefore, reveal that the initial drug distribution has a huge influence on the kinetics of drug delivery.

Many experimental drug release curves are analyzed by fitting their first part to a power law. Our results suggest that the interpretation of the kinetic exponent obtained in this way (n) might be a risky business since surface phenomena and the initial drug distribution can deeply affect this exponent. For instance, Cheng et al. reported n-values ranging from 0.42 to 0.68. The greater values were attributed to anomalous transport [52], but our simulations prove that these values can also be found in the case of purely diffusive processes if the drug is concentrated in the core of the nanogel. On the other hand, the physical interpretation of the proportionality constant (k) of the power law is much easier because it is straightforwardly related to the solute diffusion coefficient (see Equation (8)). In fact, k could even provide a first estimate of this coefficient.

Along with Equations (5) and (6), Equation (8) also reveals some parameters controlling drug delivery in neutral nanogels: matrix flexibility, nanogel radius, solute radius, polymer radius, and polymer volume fraction. However, one should keep in mind that k depends on n, and, in turn, this exponent depends on surface phenomena and the initial drug distribution. Previous CG simulations of solute diffusion in flexible gels concluded that the effects of the degree of crosslinking on diffusion are negligible if the solute size is smaller than the mesh size [29], which is quite common, but the role of the degree of crosslinking in drug delivery might be important for tightly meshed networks.

Our simulations shed some light on the physical meaning of kinetic parameters, but some aspects can be improved in future works. For example, the inclusion in the model of electrostatic and/or hydrophobic interactions between drugs and polymers could lead to more realistic release curves. In any case, diffusion from nanogels can be modified by physical interactions [53]. Regarding specific interactions, the combination of CG simulations with more sophisticated computational techniques would allow us to include chemical specificity. The encapsulation/delivery efficiency for particular nanogels and drugs (in which their molecular structure must be explicitly taken into account) might be estimated with the help of all-atom simulations.

## Figures and Tables

**Figure 1 polymers-14-04760-f001:**
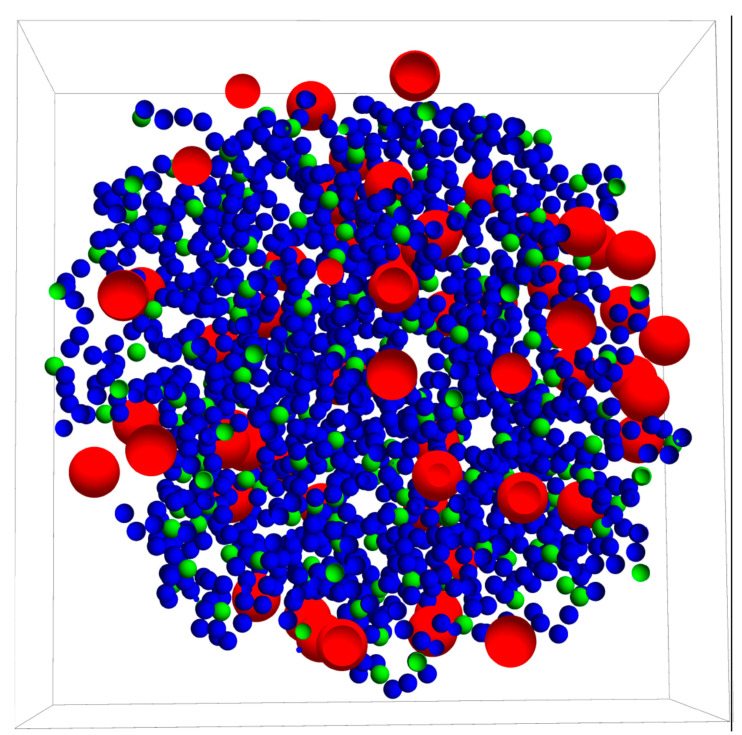
Cross-section of the nanogel and drug molecules housed within it at the initial configuration. Blue, green, and red beads represent monomeric units, crosslinkers, and drug molecules, respectively. The frontal plane passes through the center of mass of the nanogel. The mean distance between the crosslinker particles is 2.2 nm.

**Figure 2 polymers-14-04760-f002:**
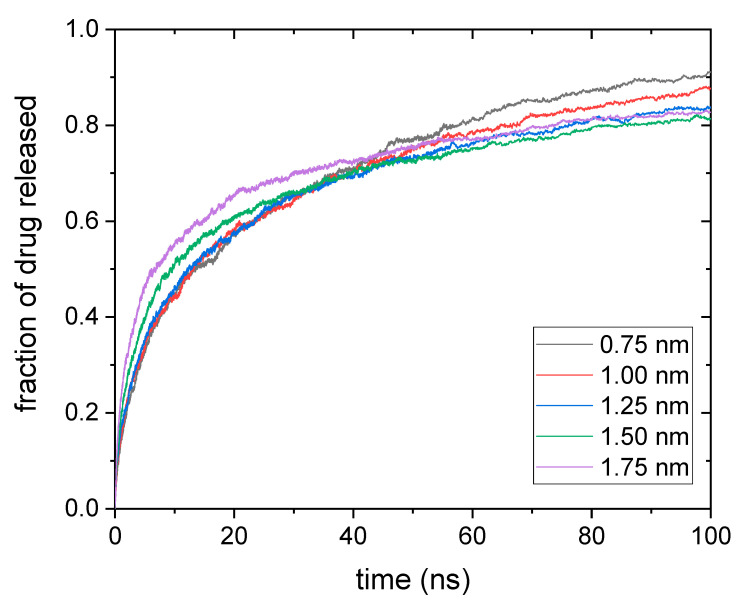
Fraction of drug released as a function of time for five solute diameters. The drug was housed in a flexible nanogel.

**Figure 3 polymers-14-04760-f003:**
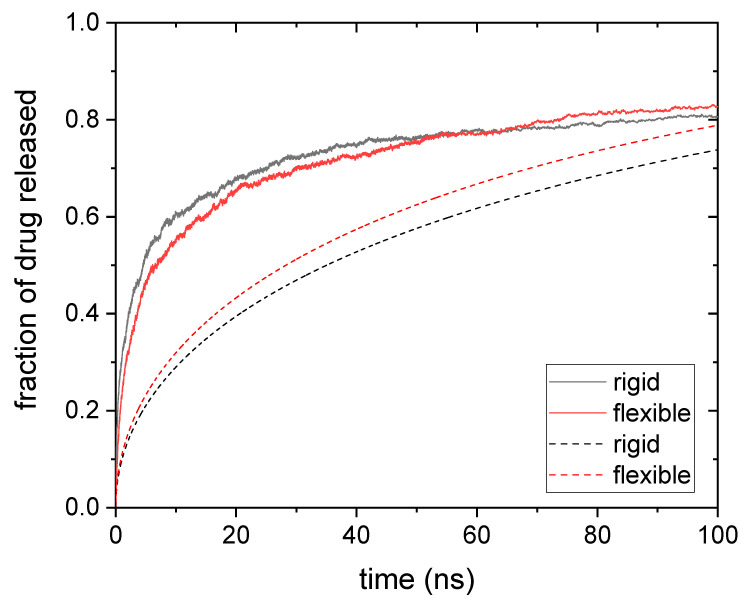
Fraction of drug released as a function of time for a drug whose diameter was 1.75 nm and was housed in rigid and flexible nanogels (black and red solid lines, respectively). The predictions obtained from Equations (4) and (5) (rigid nanogel, dashed black line) and Equations (4) and (6) (flexible nanogel, dashed red line) are also plotted.

**Figure 4 polymers-14-04760-f004:**
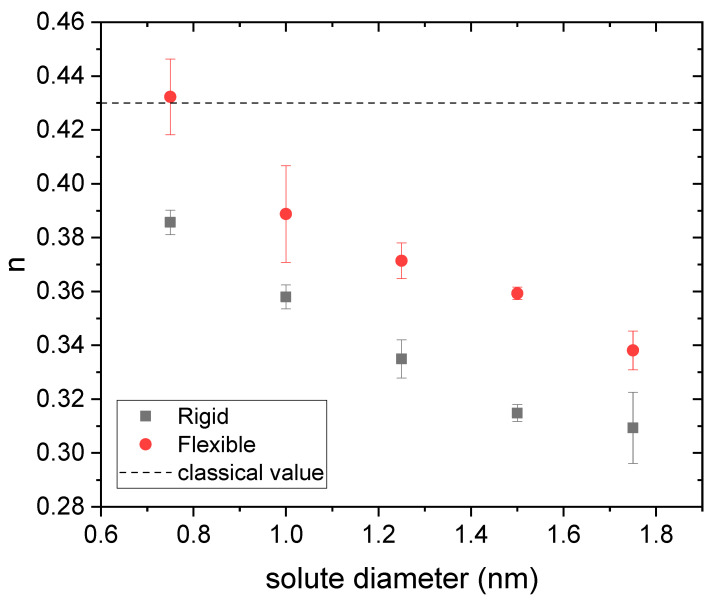
Kinetic exponent (n) obtained from three independent fits to Equation (7) as a function of the solute diameter for drugs housed in rigid and flexible nanogels (square and circles, respectively). The classical value for diffusion-controlled release is also plotted for comparison (dashed line).

**Figure 5 polymers-14-04760-f005:**
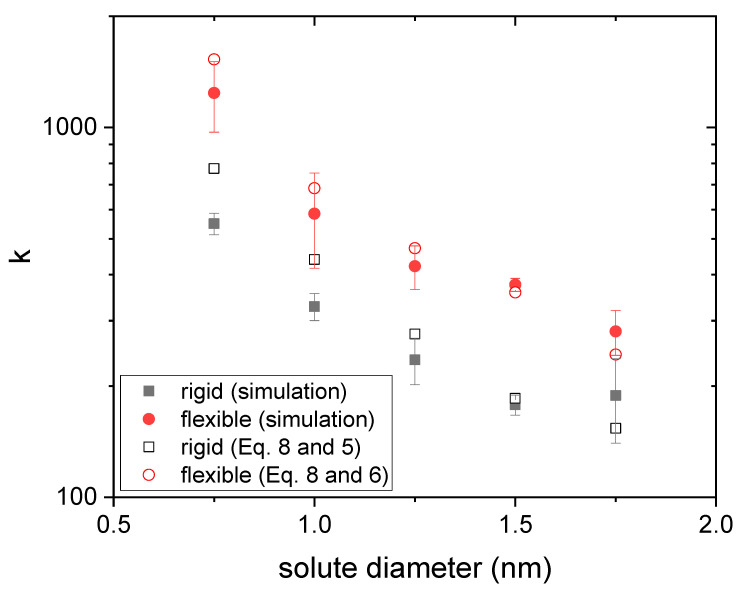
k-values obtained from three independent fits to Equation (7) as a function of the solute diameter for drugs housed in rigid and flexible nanogels (square and circles, respectively). The predictions obtained from Equations (5) and (8) (rigid nanogel, open black squares) and Equations (6) and (8) (flexible nanogel, open red circles) are also plotted.

**Figure 6 polymers-14-04760-f006:**
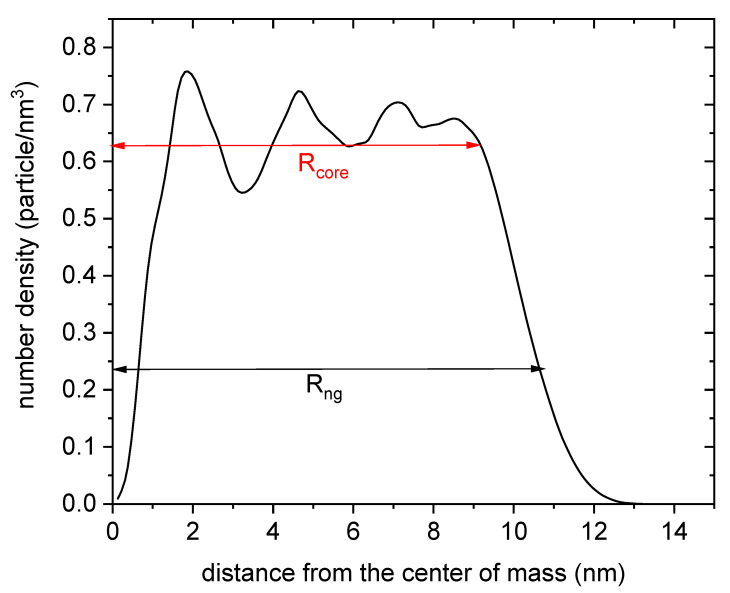
Number density of monomeric units and crosslinker molecules as a function of the distance from the center of mass of the nanogel.

**Figure 7 polymers-14-04760-f007:**
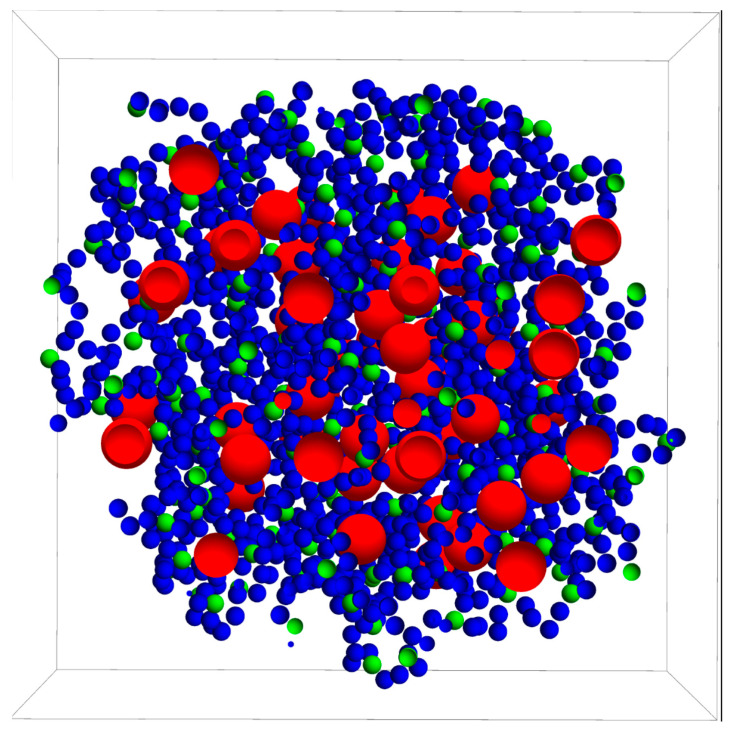
Cross-section of the nanogel and drug molecules housed only within its core at the initial configuration. Blue, green, and red beads represent monomeric units, crosslinkers, and drug molecules, respectively. The frontal plane passes through the center of mass of the nanogel. The mean distance between the crosslinker particles is 2.2 nm.

**Figure 8 polymers-14-04760-f008:**
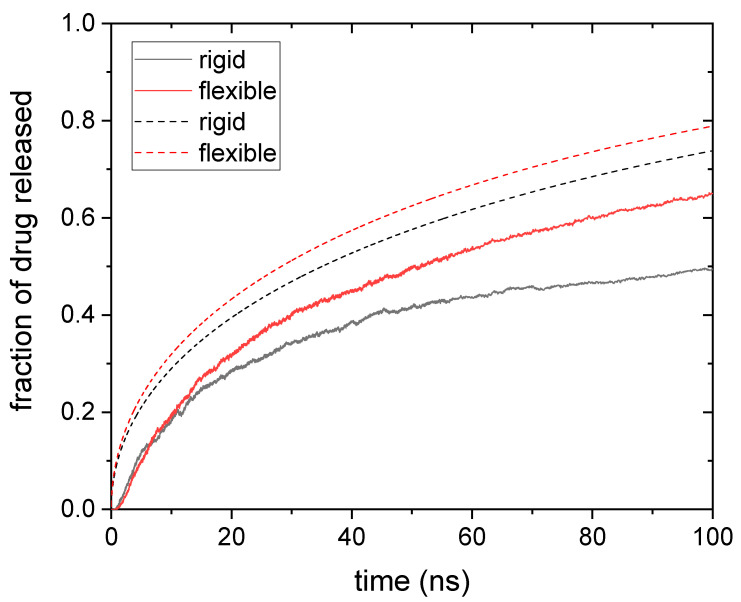
Fraction of drug released as a function of time for a drug whose diameter is 1.75 nm and is housed in the core of rigid and flexible nanogels (black and red solid lines, respectively). The predictions obtained from Equations (4) and (5) (rigid nanogel, dashed black line) and Equations (4) and (6) (flexible nanogel, dashed red line) are also plotted.

**Figure 9 polymers-14-04760-f009:**
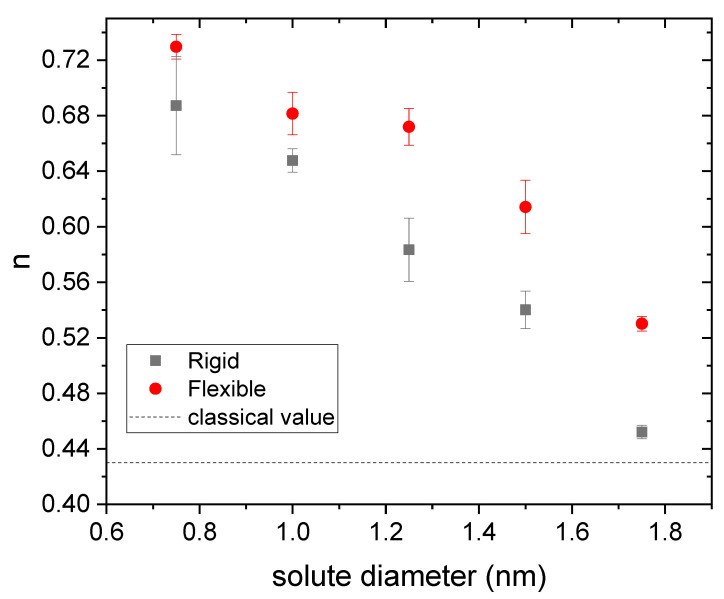
Kinetic exponent (n) obtained from three independent fits to Equation (9) as a function of the solute diameter for drugs housed in the core of rigid and flexible nanogels (square and circles, respectively). The classical value for diffusion-controlled release is also plotted for comparison (dashed line).

**Figure 10 polymers-14-04760-f010:**
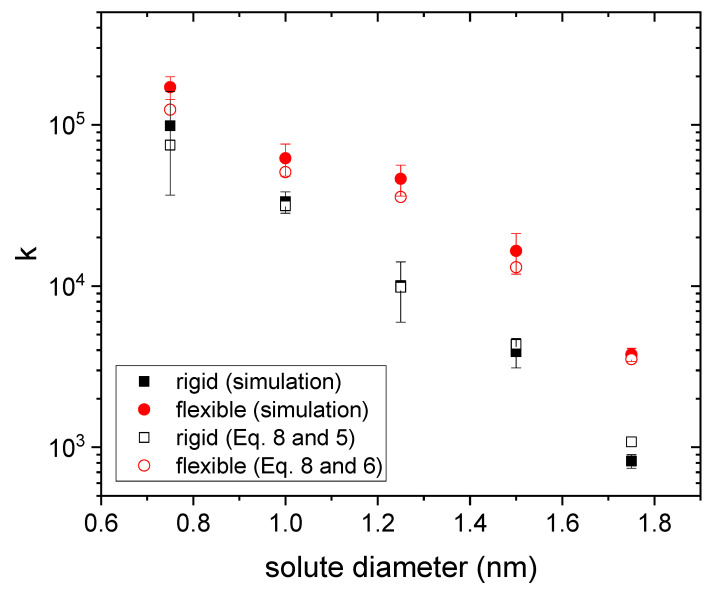
k-values obtained from three independent fits to Equation (7) as a function of the solute diameter for drugs housed in the core of rigid and flexible nanogels (square and circles, respectively). The predictions obtained from Equations (5) and (8) (rigid nanogel, open black squares) and Equations (6) and (8) (flexible nanogel, open red circles) are also plotted.

**Figure 11 polymers-14-04760-f011:**
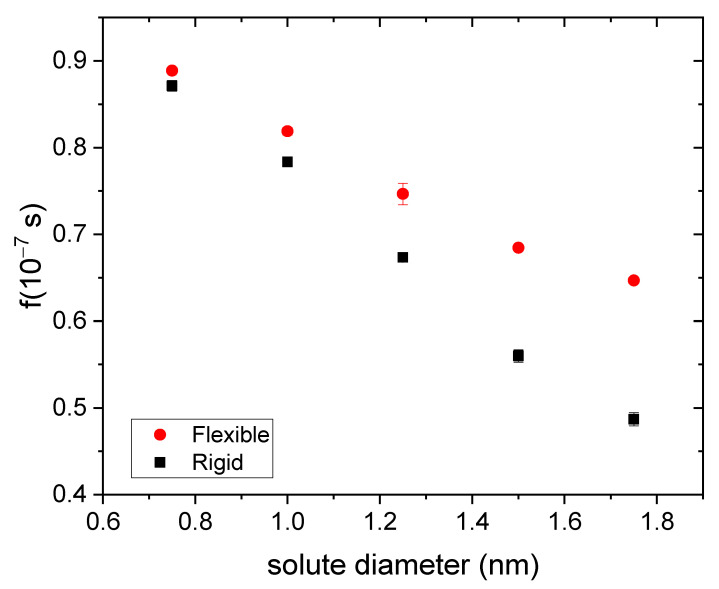
Fraction of drug released at 100 ns as a function of the solute size for rigid and flexible nanogels (square and circles, respectively). Drug molecules are housed in the core of the nanogel.

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
