# Peer review of "Coarse-Grained Simulations of Release of Drugs Housed in Flexible Nanogels: New Insights into Kinetic Parameters"

_polymers, 2022, doi:10.3390/polym14214760_

Round 1

Reviewer 1 Report

1.     In expressions (2) and (7,9), different parameters are denoted by the same letter k.

2.     In conclusion, it should be indicated what changes in the model can increase the reliability of the results obtained, as well as indicate directions for further research in this area.

Author Response

See file attached

Reviewer 2 Report

The authors present a systematic study towards understanding the effect of polymer-drug structure and topology in controlled drug release for nanogels. Several useful correlations between nanogel topology, size and distribution of drug molecules on the fraction of drug released have been provided that could help the design of an efficient nanogel drug carrier. A few questions/comments if addressed would strengthen the scope of the paper:

1.     Can the authors comment on how their measured their drug release (%f) in simulations – What defines the release of drug molecules such as Mean Square Displacement (MSD) quantity and what is the criteria to determine if the molecule has left the nanogel into the bulk solvent?

2.     a) Can the authors compare their results with experimental data (qualitatively/quantitatively) in the manuscript and provide a prediction of what they think is an important design parameter that could control the drug delivery in these nanogels? (For instance, the mesh/solute size and distribution, polymer chain stiffness/architecture recommended to achieve an effective controlled delivery of drugs)

b) Line 217-218 – The authors mention that their model does not account for polymer-drug interactions and hence could result in different drug release times/behavior compared to solutions. Can the authors comment on if a future study can account for this and if not, how their results can still apply to drug molecules and polymers with varying chemical specificity?

3.     It would be helpful if the authors could illustrate the cross-linker molecules in their visual images (Example: Figure 1). Could the authors explain how they chose the ratio of cross-linker particles and what the cross-link density is for this system. Also, how this parameter is expected to influence the drug release both at the surface and the core? 

4.     Can the authors provide a brief computational detail of how they modeled a rigid nanogel system – for instance, if the force constant was made stiffer in the rigid ones compared to flexible nanogels?

5.     Figure 5 and line 261 – Can the authors explain why the results between simulations and those predicted from theory show differ in the k values more for low solute diameter values (up to 1 nm primarily)?

6.     Line 308 – Can the authors elaborate why their system depicted in Figure 7 does not have a uniform particle distribution? If the surface effects are removed, their system is expected to be closer to the one predicted from theory albeit the differences in the perfect sink conditions assumed in the theory. If their distribution is not uniform, have they performed multiple runs to sample the heterogeneities for this system (Figure 7)?  

7.     Lines 204-206 – It is unclear from the authors’ explanation as to why the rigid nanogel would allow for more surface anisotropic diffusion (60% of drug release). One would expect that at the surface as well, the flexibility of the polymer would help in overcoming steric hindrance in an otherwise rigid surface or the differences are minimal between the rigid and flexible nanogels. Can the authors support their explanation and findings by calculating the fraction of drug molecules at the surface versus the core and perhaps the mesh size/solvent-accessible surface area/MSD or equivalent quantity to explain this behavior? 

Author Response

See file attached.

Reviewer 3 Report

This gives results of a study on release dynamics of flexible versus rigid nanogels. As such it has results which would appear to be publishable. However: 1. The actual results depend heavily upon results of their previous paper [37]. More of a review of that result would be welcome, especially upon the use of the Cichocki-Hinsen algorithm and what is meant by the m-component. 2. It is not clear how the figures 1 and 7 (nanogel cross sections) were obtained. 3. Comments are given on the small time scales of the simulations (extending to 10^(-7) Seconds. But that discussion does not seem to justify this extremely small time. Maybe more should be said about what should be done to bring the simulated times into physically realistic ones. 4. The introduction of delay has inverted the dashed versus solid lines of Figs. 4 & 9. The explanation is somewhat reasonable concerning the slope, etc. but does not seem to explain this inversion? 5. Overall this paper raises a number of questions on simulation of drug release. More on what these questions are and their importance could be discussed in the conclusions. For example why does the curve of release of Fig. 3, solid, not appear to go to a full release? And how does one evaluate the risk involved in the "risky business" mentioned? 6.. The programs used should be made available to a reader with a statement as to how to obtain it.

Author Response

See file attached

Reviewer 4 Report

1. All the equations should be centered in the text.

2. All the figures should be centered in the text.

3. you should be consistent with the style of the references, Ref. 6, 17, and 37 should be modified (use the same style as other references)

Author Response

See file attached

Round 2

Reviewer 2 Report

The authors have made a valuable contribution towards understanding the impact the nanogel particle topology and drug particle distribution towards drug release. They have also addressed the previous comments in detail and have revised the manuscript to enhance the readability and address several points. The manuscript can be accepted in its present form with minor suggested changes below:

1. Figure 11 - The authors need to mention if the fraction of drug release plotted was done only the first or the second simulated case (ie. when the drug was housed within the entire gel or the core, or both) both in the figure and text.

2. Figure 7: Similar to Figure 1, the authors need to mention if this was the initial configuration for this system

3. Figure 1/7: The authors could illustrate visually in the image or mention in the legend that the mean distance between the cross-linker particles is 2.2 nm. It will help the readers notice and appreciate that is there is spatial ordering in these systems consistent with the text.

4. The authors could mention an estimate from Figure 6 as to what fraction/percent of the drug particles are located on the surface to highlight and emphasize one of their important conclusions that surface anisotropic diffusion alters the drug release curve significantly - it would help give a sense of how even a small fraction of surface particles could change the drug release kinetics significantly.

Author Response

Dear Reviewer:

Your painstaking reading of the manuscript and suggestions are again gratefully acknowledged. Below you can find the point-by-point reply to them. Changes are marked in red.

  1. Figure 11 displays results corresponding to drug housed in the core. This is mentioned both in the figure and text, as suggested.
  2. Certainly, Figure 7 shows the initial configuration of the system. This is mentioned in its legend, as suggested.
  3. The legends of Figures 1 and 7 state that mean distance between the cross-linker particles is 2.2 nm, as recommended.
  4. An estimate of the fraction of drug molecules located on the surface is mentioned in page 7. This fraction is not small due to the high surface-to-volume ratio of nanogels. Obviously, the presence of many solute particles at the surface enhances the effects of anisotropic diffusion.
